# Early Identification and Diagnostic Approach in Acute Respiratory Distress Syndrome (ARDS)

**DOI:** 10.3390/diagnostics11122307

**Published:** 2021-12-08

**Authors:** François Arrivé, Rémi Coudroy, Arnaud W. Thille

**Affiliations:** 1Centre Hospitalier Universitaire de Poitiers, Médecine Intensive Réanimation, 86021 Poitiers, France; francois.arrive@chu-poitiers.fr (F.A.); r.coudroy@yahoo.fr (R.C.); 2Centre d’Investigation Clinique 1402 ALIVE, INSERM, Université de Poitiers, 86000 Poitiers, France

**Keywords:** intensive care unit, acute respiratory distress syndrome, acute respiratory failure

## Abstract

Acute respiratory distress syndrome (ARDS) is a life-threatening condition defined by the acute onset of severe hypoxemia with bilateral pulmonary infiltrates, in the absence of a predominant cardiac involvement. Whereas the current Berlin definition was proposed in 2012 and mainly focused on intubated patients under invasive mechanical ventilation, the recent COVID-19 pandemic has highlighted the need for a more comprehensive definition of ARDS including patients treated with noninvasive oxygenation strategies, especially high-flow nasal oxygen therapy, and fulfilling all other diagnostic criteria. Early identification of ARDS in patients breathing spontaneously may allow assessment of earlier initiation of pharmacological and non-pharmacological treatments. In the same way, accurate identification of the ARDS etiology is obviously of paramount importance for early initiation of adequate treatment. The precise underlying etiological diagnostic (bacterial, viral, fungal, immune, malignant, drug-induced, etc.) as well as the diagnostic approach have been understudied in the literature. To date, no clinical practice guidelines have recommended structured diagnostic work-up in ARDS patients. In addition to lung-protective ventilation with the aim of preventing worsening lung injury, specific treatment of the underlying cause has a central role to improve outcomes. In this review, we discuss early identification of ARDS in non-intubated patients breathing spontaneously and propose a structured diagnosis work-up.

## 1. Introduction

Acute respiratory distress syndrome (ARDS) was reported for the first time in 1967 in 12 patients with sudden respiratory failure due to a non-cardiogenic pulmonary edema [1]. None of these patients had underlying pulmonary disease, and they rapidly developed acute hypoxemia, stiff lungs, and diffuse bilateral alveolar infiltrates on chest radiograph a few days after a precipitating factor. Autopsy was performed in the 7 deceased patients and all but one had a characteristic histological pattern of diffuse alveolar damage including hyaline membranes, edema, cell necrosis and proliferation [2]. Chronology of histological lesions is characterized by an early exudative phase with intra-alveolar edema due to altered permeability of the alveolar-capillary membrane, followed by a proliferative phase of repair with intense proliferation of alveolar type-2 cells and fibroblasts, leading to normal tissue resolution or progress toward fibrosis [3]. Although diffuse alveolar damage is usually considered as the morphological hallmark of ARDS lungs, this typical pattern is actually found in less than half of the patients with clinical criteria for ARDS [4,5]. ARDS is caused by numerous etiologies of pulmonary as well as extra-pulmonary origin (Table 1). Whatever the underlying cause or the underlying histological lesion, diffuse alveolar edema and alveolar collapse decrease the available lung volume for ventilation and consequently lung compliance which, alongside microcirculation impairment, lead to ventilation-perfusion imbalances with intrapulmonary shunt and dead space [6]. In addition to lung-protective ventilation aimed at mitigating lung injury, accurate identification of the ARDS etiology is obviously of paramount importance for initiation of adequate treatment [7]. The diagnostic approach has not been widely studied in the extensive literature on ARDS. In the largest epidemiological study performed to date on ARDS (the LUNG SAFE study) [8], whereas pneumonia was reported as the main reason of ARDS, no bacterial or viral documentation was mentioned, and exams such as lung computed tomography or broncho-alveolar lavage were uncommon, even in patients without risk factor for ARDS [9]. Given the large heterogeneity of risk factors triggering ARDS, a systematic diagnostic approach should be proposed so as to accurately identify the cause of ARDS, as soon as evidence for community-acquired pneumonia has not been found during first-line diagnosis.

COVID-19 was the main reason for ARDS during recent months and was not in itself a diagnosis issue. However, the huge influx of patients in intensive care units (ICUs) during the COVID-19 pandemic has led intensivists to treat a large number of patients with noninvasive oxygenation strategies outside ICUs due to the limited number of beds available [10]. The current definition of ARDS focuses mainly on intubated patients under invasive mechanical ventilation and does not determine whether these patients treated without positive pressure ventilation meet the criteria for ARDS. A revision of this definition is probably needed with aimed at determining whether patients breathing spontaneously with noninvasive oxygenation strategies meet criteria for ARDS [11]. Earlier identification of ARDS before intubation may potentially change therapeutic strategies and future research.

## 2. Definition of ARDS

The first clinical definition of ARDS was proposed by an international American–European Consensus Conference in 1994 [12], and used the four following criteria: [1] acute onset of hypoxemia, [2] PaO_2_ to FiO_2_ ratio ≤200 mm Hg, [3] bilateral infiltrates on chest radiograph, and [4] pulmonary artery wedge pressure ≤18 mm Hg or no clinical sign of left atrial hypertension [12]. Patients meeting all these criteria but having less severe hypoxemia with a PaO_2_/FiO_2_ ratio between 201 and 300 mmHg were considered as having acute lung injury, not ARDS. Whereas these clinical criteria for ARDS have been widely adopted by clinicians and researchers, their application has raised some issues over time and this definition has been criticized on each criterion [13], leading in 2012 to a new definition, the Berlin definition [14].

The changes proposed in the Berlin definition addressed most of the previous definition limitations. First, the “acute onset” of ARDS was specified, and respiratory symptoms must appear or worsen within 7 days of a clinical insult. Timing accuracy enables exclusion of patients who develop respiratory failure over several weeks such as idiopathic pulmonary fibrosis, nonspecific interstitial pneumonitis, bronchiolitis obliterans with organizing pneumonia, or granulomatosis with polyangiitis [15]. Second, severity has been classified according to hypoxemia: mild when PaO_2_/FiO_2_ ratio is between 201 and 300 mmHg, moderate when PaO_2_/FiO_2_ ratio is between 101 and 200 mmHg, and severe when PaO_2_/FiO_2_ ratio is equal to or below 100 mmHg [14]. Oxygenation criteria were well-correlated to severity with mortality increasing from 27% in mild, to 32% in moderate, and up to 45% in severe ARDS. As a major limitation of the previous definition was assessment of PaO_2_/FiO_2_ ratio regardless of the positive pressure used [16,17], the Berlin definition stated that PaO_2_/FiO_2_ ratio had to be measured with a positive end-expiratory pressure (PEEP) level of at least 5 cmH_2_O [14]. Third, as high values of pulmonary wedge pressure are commonly observed in patients with ARDS [18,19] and since the routine use of pulmonary artery catheter is pointless for hemodynamic management [20], the Berlin definition stated that respiratory failure must not be fully explained by cardiac failure of fluid overload as judged by the clinician or confirmed by echocardiography. Fourth, as chest radiographs have only moderate inter-observer reliability [21,22], the Berlin definition considered radiological findings as bilateral opacities not only on chest X-ray, but also on CT-scan. Using the Berlin definition, the large international observational LUNG SAFE study reported that ARDS represented around 10% of ICU admissions and 23% of patients requiring invasive mechanical ventilation [8]. This represented at least 5 patients per bed and per year. Overall hospital mortality was 40%, and up to 46% for severe ARDS. However, this study highlighted the poor recognition of ARDS criteria by clinicians, especially for mild forms that were recognized in only half of the patients, depriving them of access to effective secondary prevention measures [23]. Early recognition of ARDS in these less severely hypoxemic patients appears to be an important issue, insofar as half of them worsen in severity during the first week, with high mortality [24].

Despite these changes, numerous limitations persist in the Berlin definition. First, even though adjustment of PEEP level and FiO_2_ have a major influence on oxygenation, ARDS severity is assessed on a single blood gas measurement without prior standardized ventilator settings. Oxygenation is always better with high than with lower PEEP levels [25,26,27,28]. After increasing PEEP, several studies have shown that a high proportion of patients had their category of severity modified, from severe to moderate or mild, or from moderate to mild [29,30,31]. FiO_2_ variations may also be associated with significant changes in PaO_2_/FiO_2_ ratio [17,29], and it has been shown that, for the same PaO_2_/FiO_2_ ratio, patients ventilated with high FiO_2_ had higher mortality than those ventilated with lower FiO_2_ [16]. Likewise, time from optimization of ventilator settings to PaO_2_/FiO_2_ measurement seems crucial. Indeed, Villar and colleagues reported that mortality was more reliably predicted according to the three categories of severity when PaO_2_/FiO_2_ ratio was measured with a PEEP level of at least 10 cm H_2_O, and with FiO_2_ level of at least 50% [30,31]. Moreover, assessment of PaO_2_/FiO_2_ ratio yielded a more clinically relevant ARDS classification when measured 24 h after ARDS onset, and therefore, standardized ventilator settings with a PEEP level of at least 10 cm H_2_O and the persistence of hypoxemia may help to improve ARDS classification. Second, assessment of lung opacity extension using chest radiographs remains a major limitation. ARDS can encompass patients with lobar opacities involving only the 2 lower quadrants (focal ARDS) as well as patients with diffuse opacities involving the 4 quadrants (diffuse ARDS). However, the most adequate strategy of mechanical ventilation could be different according to lung morphology, especially including higher PEEP levels in patients with diffuse opacities than in those with lobar opacities [32,33]. Whereas personalized mechanical ventilation tailored to lung morphology may improve outcomes, it has been shown that patients are frequently misclassified as diffuse or focal when assessment is based on chest radiographs alone [33]. Diffuse opacities involving the four quadrants also seem to be a strong marker of diffuse alveolar damage [3]. Whereas extension of opacities may impact therapeutics and outcomes [34], it often remains poorly assessed in clinical practice. Lastly, up until the Berlin definition, ARDS was considered only in patients intubated under invasive mechanical ventilation. As a footnote, the Berlin definition specified that patients receiving noninvasive continuous positive airway pressure of at least 5 cm H_2_O may be considered as mild ARDS cases. It is disappointing that the issue of patients treated noninvasively was not addressed in the Berlin definition, even though noninvasive ventilation was being increasingly used for management of ARDS. A few years later, the LUNG SAFE study showed that 15% of ARDS cases were managed with noninvasive ventilation and that classification of ARDS severity based on PaO_2_/FiO_2_ measured under noninvasive ventilation was well- correlated to the risk of intubation and mortality [35]. These findings showed that patients treated with noninvasive ventilation could be considered moderate or even severe, as well as mild ARDS. At the same time, while high-flow nasal cannula oxygen therapy has grown worldwide in management of acute respiratory failure [36], these patients still cannot be considered as ARDS according to the Berlin definition, since their positive airway pressure remains lower than 5 cm H_2_O. Given all these limitations, it is high time to revise the current definition to determine whether patients with diffuse opacities and breathing spontaneously under noninvasive oxygenation strategies could be considered at an earlier stage in the course of respiratory failure as ARDS cases.

## 3. Early Identification of ARDS in Patients Breathing Spontaneously

The huge influx of ICU patients during the COVID-19 pandemic has led intensivists to treat a large number of patients with noninvasive oxygenation strategies due to ventilator shortage [37], and even outside ICUs due to the limited number of available beds [10]. These patients were treated in the ward with high-flow nasal oxygen, continuous positive airway pressure or noninvasive ventilation and only those requiring intubation were admitted to an ICU. The current definition of ARDS focuses only on intubated patients under invasive mechanical ventilation and does not consider patients breathing spontaneously as having ARDS, even if they meet all the other clinical criteria for ARDS. Indeed, according to the Berlin definition, a minimal PEEP level of 5 cm H_2_O is needed to meet clinical criteria for ARDS, which is usually not reached using high-flow nasal oxygen, even with a flow at least 50 L/min [38,39]. And even if the Berlin definition considers patients treated with noninvasive ventilation as having ARDS, they can only be considered as mild ARDS. However, the greater the severity of hypoxemia under noninvasive ventilation, the higher the risk of intubation and mortality [35,40,41,42]. The rates of intubation for patients with acute respiratory failure treated in ICUs with noninvasive ventilation or high-flow nasal oxygen range from 35 to 55% [35,36,40,41], meaning that a high proportion of patients will be considered as ARDS patients once invasive mechanical ventilation is initiated, and at least those with persistent hypoxemia and bilateral infiltrates. Besides the fact that these patients will be belatedly identified as ARDS cases, we cannot rule out the possibility that those successfully treated with noninvasive oxygenation strategies had ARDS and could have benefited from the same pharmacological treatments.

Patients could even be identified at an earlier stage as having ARDS while breathing spontaneously under standard oxygen. In a study pooling 219 ICU patients treated first with standard oxygen and then with noninvasive ventilation for acute hypoxemic respiratory failure, 180 (82%) had bilateral infiltrates [43]. Among those with PaO_2_/FiO_2_ ≤ 300 mm Hg under standard oxygen, 94% fulfilled criteria for ARDS once NIV was applied with a PEEP level of at least 5 cm H_2_O, meaning that almost all patients admitted to ICU with pulmonary bilateral infiltrates and a PaO_2_/FiO_2_ ≤ 300 mm Hg under standard oxygen meet ARDS criteria. Although PaO_2_/FiO_2_ was significantly higher and the proportion of patients with severe hypoxemia significantly lower with NIV than with standard oxygen, few patients increased their PaO_2_/FiO_2_ above 300 mm Hg and therefore almost all of them still met the ARDS criteria. In this study, in-ICU mortality rate was 29% in patients with bilateral infiltrates and PaO_2_/FiO_2_ less than or equal to 300 mm Hg under standard oxygen [43], a rate very close to that of intubated patients with ARDS according to the Berlin definition (30%) [14], and only slightly lower than the 35% reported in intubated patients included in the LUNG SAFE study [8]. To accurately estimate FiO_2_ in patients breathing spontaneously with a reservoir bag mask, the 3% formula (FiO_2_ estimated = 3% per L of oxygen + 21%) seems optimal [44].

High-flow nasal cannula oxygen therapy has been widely used for management of respiratory failure due to COVID-19 [37,45,46,47,48,49]. In 2015, a randomized controlled trial showed for the first time that high-flow nasal oxygen (with a flow of 50 L/min) may reduce mortality of patients with acute hypoxemic respiratory failure as compared to standard oxygen or noninvasive ventilation using face mask [36]. Although high-flow nasal oxygen may generate continuous positive airway pressure above 5 cm H_2_O with a particularly high flow exceeding 50 L/min and with mouth closed, PEEP levels are often 2–3 cm H_2_O and remain lower than the levels needed to reach ARDS criteria [38,39]. Nonetheless, it has been shown that biomarkers of inflammation and injury in patients with bilateral infiltrates and acute respiratory failure treated by high-flow nasal oxygen increased to values similar to those of ARDS patients under invasive mechanical ventilation [50]. During the COVID-19 pandemic, the rates of intubation were particularly high in patients treated with high-flow nasal oxygen as first-line oxygenation strategy [37,45,46,47,48,49], meaning that these patients could have been considered as ARDS cases as soon as high-flow nasal oxygen was initiated. Besides, a new definition for ARDS including patients treated with high-flow nasal oxygen set at a flow at least 30 L/min has recently been proposed, taking as a given fact that patients with COVID-19 actually have ARDS [51].

Early identification of ARDS is likely to be a major issue in assessment of future pharmacological and non-pharmacological treatments. Numerous anti-inflammatory drugs have been unsuccessfully assessed in ARDS. Steroids were initially evaluated at a late stage in the course of ARDS with the aim of preventing evolution toward fibrosis [52,53,54]. However, mortality was significantly higher when steroids were started after 2 weeks of evolution [54]. By contrast, several studies in which steroids were started early in the course of ARDS, i.e., within the first 3 days after intubation, have shown beneficial effects on outcomes with a decreased risk of death [55,56]. During the COVID-19 pandemic, it was shown that steroids significantly reduced the risk of death not only in patients under invasive mechanical ventilation, but also in patients breathing spontaneously under standard oxygen [57]. COVID-19 is characterized by a marked systemic inflammatory response and in this setting, steroids may be particularly effective. However, it cannot be ruled out that steroids could be beneficial in all forms of ARDS when initiated at an early stage [58]. Early identification of ARDS would allow assessment of anti-inflammatory drugs or future pharmacological treatments at a much earlier stage of the disease. Recognition of ARDS at an early stage might also be of paramount importance to initiate lung-protective measures in patients breathing spontaneously with a high respiratory drive. Indeed, an emerging concept, known as patient self-inflicted lung injury, deals with the risk of progression of lung injury in patients with respiratory failure generating large tidal volumes and subsequent high transpulmonary pressures [59]. Similarly, prone positioning has demonstrated its efficacy in ARDS treatment amongst invasively ventilated patients [60], and was recently proposed, with promising results regarding the risk of intubation in COVID-19 patients undergoing high-flow nasal oxygen [61]. Therefore, early identification of ARDS in patients breathing spontaneously and not yet intubated could be useful as well for early initiation of anti-inflammatory drugs as for lung-protective measures. We believe that the future definition could consider patients treated with at least 30 L/min of high-flow oxygen or with noninvasive ventilation as in fact having ARDS and allow to consider severity as mild, moderate or severe ARDS according to their PaO_2_/FiO_2_ measured under this noninvasive oxygenation strategy, and not only as mild ARDS.

## 4. Proposed Diagnostic Approach for Identification of ARDS Etiology

Usual risk factors triggering ARDS include direct injury of pulmonary origin (pneumonia, aspiration, toxic inhalation, lung contusion, near-drowning and pulmonary vasculitis) and indirect injury of extra-pulmonary origin (non-pulmonary sepsis, pancreatitis, non-cardiogenic shock, major trauma, blood transfusion, drug overdose, severe burns) (Table 1) [12]. In the LUNG-SAFE study, pneumonia was the most common risk factor representing approximately 60% of all ARDS cases [8]. However, pulmonary infection was not documented, and no data were provided concerning the bacteria, viral, fungal or other origin of pneumonia.

In case of pulmonary ARDS, a first-line diagnostic work-up should include a detailed and complete search for any underlying pulmonary infection. Viral pulmonary infection can easily be identified using RT-PCR (SARS-CoV2 and seasonal Coronavirus, Influenza, Parainfluenza, Rhinovirus, Respiratory syncytial virus, Human Metapneumovirus, Adenovirus, Bocavirus, and Enterovirus) from nasal swab in non-intubated patients or from tracheal aspirates in intubated patients. Alongside standard microbiology on tracheal aspirates in intubated patients, blood culture, and urine antigens (*Streptococcus pneumoniae* and *Legionella pneumophila*) as recommended in diagnostic strategy for severe pneumonia [62], new molecular tools could improve the accuracy of bacterial diagnosis, and help clinicians to make antibiotic de-escalation decisions [63,64].

To date, no clinical practice guidelines have recommended structured diagnostic work-up in ARDS patients. In the LUNG SAFE study, lung CT-scan, bronchoalveolar lavage or auto-immunity tests were seldom performed, even in ARDS patients without documented diagnostic [8]. In the absence of obvious diagnosis and in case of non-documented pneumonia, we propose here a second-line diagnostic work-up (Table 2). Some less common micro-organisms responsible for atypical pneumonia may be diagnosed using molecular tools. Underlying immunocompromised status, especially any cancer, hematological disease, or human immunodeficiency virus should be suspected. Fungal pathogens mainly including *Aspergillus* spp. and *Pneumocystis jirovecii* should be considered in immunocompromised patients. Although *Aspergillus* spp. can be isolated from culture of tracheal aspirates in intubated patients, fiberoptic bronchoscopy is required to identify invasive pulmonary aspergillosis following galactomannan test and RT-PCR on bronchoalveolar lavage, while serum Beta-D-glucan antigens may help to rule out fungal infection in immunocompromised patients [65]. However, invasive pulmonary aspergillosis is increasingly diagnosed in critically ill non-immunocompromised patients, especially in those with viral pneumonia such as severe influenza [66,67,68] or COVID-19 [69,70,71], and bronchoalveolar lavage may be advisable to rule out invasive pulmonary aspergillosis in this setting. Bronchoalveolar lavage remains an essential diagnostic tool not only for diagnosis of pulmonary fungal infection in immunocompromised patients under invasive mechanical ventilation, but also for diagnosis of invasive pulmonary aspergillosis in patients with viral ARDS due to severe influenza or COVID-19. In these patients, several studies have reported an incidence of invasive pulmonary aspergillosis ranging from 15 to 30% of patients [66,67,68,69,70]. Whether galactomannan tests and RT-PCR on tracheal aspirates have the same performance as bronchoalveolar lavage for diagnosis of aspergillosis remains unknown [72,73]. 

Performing bronchoalveolar lavage in patients with acute respiratory failure while breathing spontaneously is more debatable but should be discussed according to the suspected diagnosis. Indeed, one study showed that around one-third of patients thereby increased their oxygen support, and the procedure led 15% of them to intubation [74]. A randomized controlled trial including cancer patients compared diagnosis strategy using bronchoalveolar lavage vs. noninvasive testing in patients with acute respiratory failure breathing spontaneously [75]. Whereas the need for mechanical ventilation was not significantly greater in patients who had bronchoalveolar lavage as compared to the others, noninvasive testing was not inferior to bronchoalveolar lavage as a means of identifying the cause of respiratory failure. However, galactomannan test and RT-PCR on bronchoalveolar lavage were not yet available at that time for the diagnosis of invasive pulmonary aspergillosis. Diagnostic tools have markedly improved and nowadays these tests are of paramount importance in diagnosis of invasive pulmonary aspergillosis [73]. Thereby, bronchoalveolar lavage must be considered if the main suspected diagnosis is invasive pulmonary aspergillosis, even in non-intubated patients.

In case of ARDS without usual risk factors, i.e., ARDS “mimickers”, complete diagnostic work-up including lung computed tomography, broncho-alveolar lavage and auto-immunity tests should be performed in diagnosis of ARDS etiology (Figure 1). Gibelin and colleagues showed that patients without usual risk factors represented around 7–8% of ARDS [76]. These so-called “mimickers” were divided into the following four patterns: immune, drug-induced, malignant, and idiopathic ARDS. Although these ARDS mimickers were less likely to have shock at ICU admission, they had higher mortality than patients with typical ARDS [76]. Whereas the typical time course for symptoms of idiopathic pulmonary fibrosis (usual interstitial pneumonia or nonspecific interstitial pneumonia) usually occurs over weeks or months and consequently does not meet the clinical criteria for ARDS, the mimickers may develop acute symptoms over only a few days. Pulmonary vasculitis and other auto-immune conditions can be ruled out with complete auto-immunity and hypersensitivity serological testing (Table 2). Recently, several studies have reported ARDS revealing autoimmune features such as anti-synthetase or myositis-associated antibodies in patients without extra-pulmonary manifestations [77,78,79]. However, bronchoalveolar lavage may still help in diagnosis in a number of diseases characterized by a specific alveolar inflammation cellular pattern such as increased lymphocytes in acute hypersensitivity/drug-induced ARDS, increased lymphocytes as well as increased neutrophils and eosinophils in organizing pneumonia (bronchiolitis obliterans), alveolar hemorrhage in granulomatosis with polyangiitis, or increased eosinophils in acute eosinophilic pneumonia [15]. After compete diagnostic work-up in all ARDS mimickers, Gibelin and colleagues showed that more than 75% of patients had a specific diagnosis [76]. Contrary to this study, the LUNG-SAFE study revealed that specific diagnosis was made in less than 20% of cases of ARDS mimickers [9]. However, lung CT-scan was performed in fewer than one-third of cases, bronchoalveolar lavage in fewer than 10%, and auto-immunity in only 5%, thereby underlining the need for a complete diagnostic work-up in this population.

Lastly, the role of open-lung biopsy in the work-up diagnosis of ARDS is probably limited to rare cases without diagnosis, especially when searching for cancer or organizing pneumonia. Over twenty years ago, a retrospective study about 37 open-lung biopsies in ARDS patients with negative bacterial cultures suggested that it could change therapeutic management in the majority of cases, particularly with a histologic diagnosis of cytomegalovirus pneumonia in more than half of cases [80]. These findings must be considered with caution in the era of molecular biology diagnosis since cytomegalovirus is currently more easily diagnosed using RT-PCR in bronchoalveolar lavage. In a more recent study, Guerin and colleagues showed that diffuse alveolar damage was the main histological lesion observed in open-lung biopsies for non-resolving ARDS [81]. Whereas the other patients exhibited interstitial lung fibrosis, organized pneumonia, alveolar hemorrhage or cancer, no major change in therapeutic strategy could be proposed. Consequently, the need for open-lung biopsy is nowadays uncommon and should be discussed only in highly selected patients without risk factor for ARDS and without underlying diagnosis despite complete work-up [82].

## 5. Conclusions

Expending the ARDS definition with the aim of including patients treated with noninvasive oxygenation strategies and meeting all other criteria may enable early identification of ARDS and early initiation of therapeutic strategies. A complete diagnostic work-up initially including viral and bacterial pathogens, and subsequently including lung computed tomography, bronchoalveolar lavage and auto-immunity tests is needed for the diagnosis of ARDS without evident etiology.

## Figures and Tables

**Figure 1 diagnostics-11-02307-f001:**
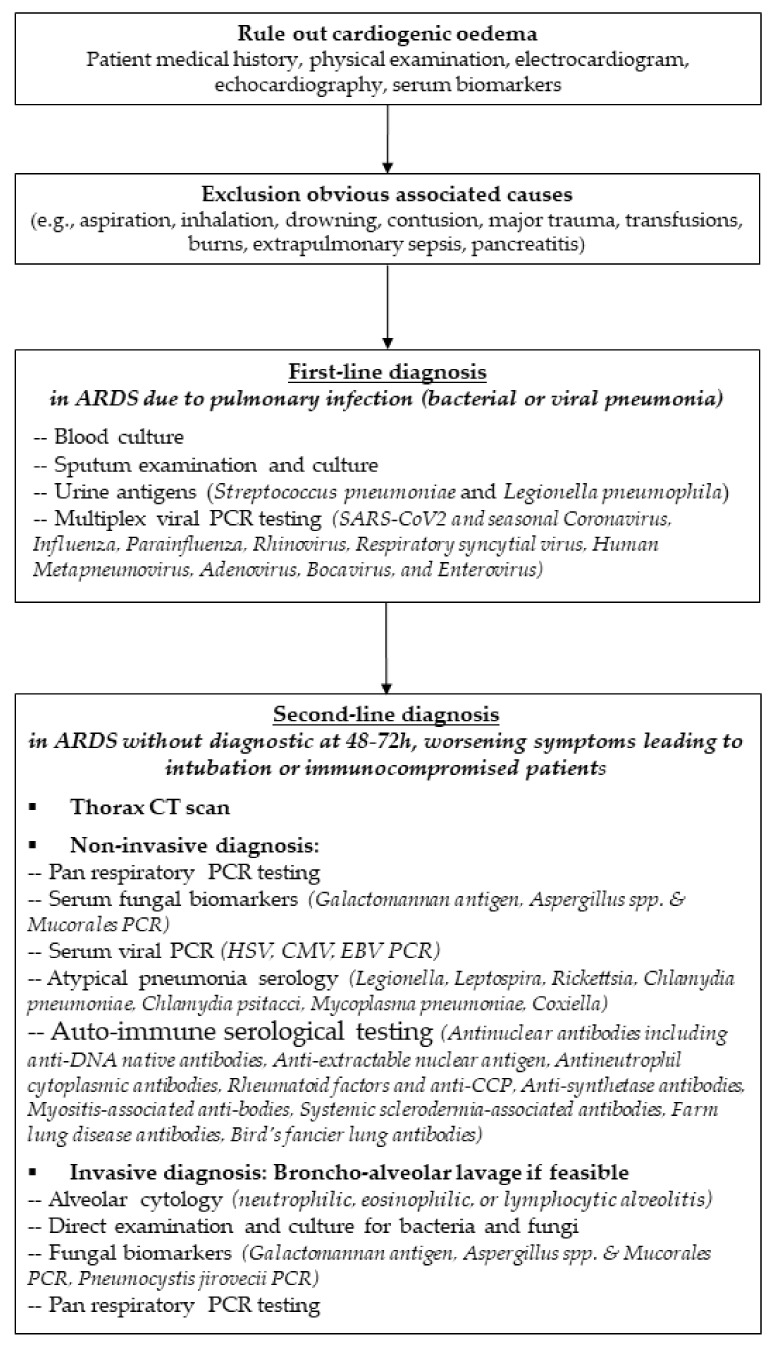
Diagnostic work-up for patients with acute respiratory distress syndrome.

**Table 1 diagnostics-11-02307-t001:** Common risk factors triggering acute respiratory distress syndrome (ARDS).

	Pulmonary ARDS (Direct Injury)	Extra-Pulmonary ARDS (Indirect Injury)
**Infectious**	- Bacterial pneumonia - Viral pneumonia - Fungal pneumonia	- Extra-pulmonary sepsis (urinary tract, abdominal, skin/soft tissue)
**Non-infectious**	- Malignant - Immune: Vasculitis or Auto-immunity - Hypersensitivity - Eosinophilic pneumonia - Aspiration of gastric contents - Inhalation injury - Pulmonary contusion - Drowning - Idiopathic	- Non-cardiogenic shock - Multiple transfusion or transfusion-related acute lung injury (TRALI) - Pancreatitis - Major trauma - Severe/extended burns - Drug-induced (systemic)

**Table 2 diagnostics-11-02307-t002:** Proposition for a second-line biological diagnostic work-up for patients with acute respiratory failure, bilateral opacities on chest X-ray and PaO_2_/FiO_2_ ratio under 300 mm Hg after exclusion of cardiogenic pulmonary edema, and no diagnosis after first-line check-up.

Microbiology	Auto-Immunity–Hypersensitivity
** *Serum* **	
- *Legionella*, *Leptospira*, *Rickettsia*, *Chlamydia pneumoniae*, *Chlamydia psitacci*, *Mycoplasma pneumoniae*, *Coxiella* serologies - HIV serology - HSV, CMV, EBV PCR - Beta-D-Glucan antigen - Galactomannan antigen - *Aspergillus* spp. & Mucorales PCR	- Antinuclear antibodies including anti-DNA native antibodies - Anti-extractable nuclear antigen (anti-ENA including anti-SM, anti-RNP, anti-SSA, anti-SSB, anti-JO-1, anti-SCL 70, anti-CENP-B) - Antineutrophil cytoplasmic antibodies (Anti-MPO, PR3) - Rheumatoid factors and anti-CCP - Anti-synthetase antibodies (anti-Jo1, PL7, PL12, EJ, OJ, KS, ZO, HA, SRP) - Myositis-associated anti-bodies (anti-Mi2, MDA5, TIF1 gamma, Ro52, SAE1, SAE2, NXP2) - Systemic sclerodermia-associated antibodies (anti-PM-Scl100, PM-Scl75, Ku, ARNpol III, TH/TO, fibrillarin) - Farm lung disease antibodies - Bird’s fancier lung antibodies - Angiotensin convertase
** *Sputum (if available and no BAL performed)* **	
- Bacterial direct examination–culture (including slow growth pathogens) - Fungal direct examination–culture - *Pneumocystis jirovecii* PCR - Pan-respiratory PCR *	
** *Broncho-Alveolar Lavage (if performed)* **	
- Cytology count and pathology examination for cytology, - Cytopathogen effect, bacteria, fungal hyphae - Direct examination and culture for bacterial and fungal pathogens (including slow-growth pathogens) - Pan-respiratory PCR * - Galactomann antigen - *Aspergillus* spp., *Mucorales*, *Pneumocystis Jirovecii* PCR - HSV, CMV, EBV PCR	- Cytology count looking for a neutrophilic, eosinophilic, or lymphocytic alveolitis) and pathology examination for hemosiderin quantification (alveolar hemorrhage).

* Pan-respiratory PCR (for example Biomerieux^®^ Biofire^®^ Filmarray^®^ Pneumonia Plus Panel: *Acinetobacter calcoaceticus-baumannii* complex, *Enterobacter cloacae*, *Escherichia coli*, *Haemophilus influenzae*, *Klebsiella aerogenes*, *Klebsiella oxytoca*, *Klebsiella pneumoniae* group, *Moraxella catarrhalis*, *Proteus* spp., *Pseudomonas aeruginosa*, *Serratia marcescens*, *Staphylococcus aureus*, *Streptococcus agalactiae*, *Streptococcus pneumoniae*, *Streptococcus pyogenes*, *Legionella pneumophila*, *Mycoplasma pneumoniae*, *Chlamydia pneumoniae*, Influenza A and B, Parainfluenza, Adenovirus, Coronavirus, Respiratory Syncytial virus, Rhinovirus/Enterovirus, Human Metapneumovirus, Middle East Respiratory Syndrome Coronavirus).

## Data Availability

Not applicable.

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
