# Peer review of "Early Identification and Diagnostic Approach in Acute Respiratory Distress Syndrome (ARDS)"

_diagnostics, 2021, doi:10.3390/diagnostics11122307_

Round 1
Reviewer 1 Report
Excellent, clinically relevant and well organised review. The point should be explicitly made that the failure to diagnose mild ARDS (50% SAFE ref 8) deprives patients access to effective secondary prevention measures.
My suggested changes are minor.
Table 1 is messy. Is gastric aspiration really infectious ab initio - I don't think so?
Page 6 final para: categorical statements are made about the use of beta-d-glucan and galactomannin but these are not supported by references from a general population. I think the assertion should be qualified.
Page 6 final para, line 267: do you mean "inadvisable"?
I think it would be worth saying that the evidence base underlying recommendation for the diagnostic work up is currently poor.
Author Response
Reviewer 1
Excellent, clinically relevant and well organised review. The point should be explicitly made that the failure to diagnose mild ARDS (50% SAFE ref 8) deprives patients access to effective secondary prevention measures.
Response:
Thank you very much for you very helpful comments.
We added in the manuscript the following:
Using the Berlin definition, the large international observational LUNG SAFE study reported that ARDS represented around 10% of ICU admissions and 23% of patients requiring invasive mechanical ventilation [8]. This represented at least 5 patients per bed and per year. Overall hospital mortality was 40%, and up to 46% for severe ARDS. However, this study highlighted the poor recognition of ARDS criteria by clinicians, especially for mild forms that were recognized in only half of the patients, depriving them of access to effective secondary prevention measures [23]. Early recognition of ARDS in these less severely hypoxemic patients appears to be an important issue, insofar as half of them worsen in severity during the first week, with high mortality [24].
My suggested changes are minor.
Table 1 is messy. Is gastric aspiration really infectious ab initio - I don't think so?
Response:
We agree with you and we reorganized the table to make it clearer.
Page 6 final para: categorical statements are made about the use of beta-d-glucan and galactomannan but these are not supported by references from a general population. I think the assertion should be qualified.
Response:
You are right and we added 2017 pulmonary aspergillosis guidelines from ESCMID/ECMM/ERS as reference for immunocompromised and non-immunocompromised hosts.
For influenzae- and COVID19-associated pulmonary aspergillosis in the ICU, references 66 and 73 are the most recent up-to-date guidelines or expert opinion for diagnosis and treatment.
This paragraph was a little bit confusing and has been rewritten to make it clearer as follows:
Although Aspergillus spp. can be isolated from culture of tracheal aspirates in intubated patients, fiberoptic bronchoscopy is required to identify invasive pulmonary aspergillosis following galactomannan test and RT-PCR on bronchoalveolar lavage, while serum Be-ta-D-glucan antigens may help to rule out fungal infection in immunocompromised patients [add ref PMID: 29544767]. However, invasive pulmonary aspergillosis is increasingly diagnosed in critically ill non-immunocompromised patients, especially in those with viral pneumonia such as severe influenza [66–68] or COVID-19 [69,70], and bronchoalveolar lavage may be advisable to rule out invasive pulmonary aspergillosis in this setting. Bronchoalveolar lavage remains an essential diagnostic tool not only for diagnosis of pulmonary fungal infection in immunocompromised patients under invasive mechanical ventilation, but also for diagnosis of invasive pulmonary aspergillosis in patients with viral ARDS due to severe influenza or COVID-19. In these patients, several studies have reported an incidence of invasive pulmonary aspergillosis ranging from 15 to 30% of patients [66–68][69,70]. Whether galactomannan tests and RT-PCR on tracheal aspirates have the same performance as bronchoalveolar lavage for diagnosis of aspergillosis re-mains unknown [73,74]. Performing bronchoalveolar lavage in patients with acute respiratory failure while breathing spontaneously is more debatable but should be discussed according to the suspected diagnosis. Indeed, one study showed that around one-third of patients thereby in-creased their oxygen support, and the procedure led 15% of them to intubation [71]. A randomized controlled trial including cancer patients compared diagnosis strategy using bronchoalveolar lavage vs. noninvasive testing in patients with acute respiratory failure breathing spontaneously [72]. Whereas the need for mechanical ventilation was not significantly greater in patients who had bronchoalveolar lavage as compared to the others, noninvasive testing was not inferior to bronchoalveolar lavage as a means of identifying the cause of respiratory failure. However, galactomannan test and RT-PCR on bronchoalveolar lavage were not yet available at that time for the diagnosis of invasive pulmonary aspergillosis. Diagnostic tools have markedly improved and nowadays these tests are of paramount importance in diagnosis of invasive pulmonary aspergillosis [73]. Thereby, bronchoalveolar lavage must be considered if the main suspected diagnosis is invasive pulmonary aspergillosis, even in non-intubated patients.
Page 6 final para, line 267: do you mean "inadvisable"?
Response:
As above-mentioned, we rewrote this paragraph to make it clearer.
I think it would be worth saying that the evidence base underlying recommendation for the diagnostic work up is currently poor
Response:
We fully agree with you and to date, unfortunately, no recommendation is available for the diagnostic work-up in patients with ARDS.
In accordance with the reviewer, we added in the manuscript:
To date, no clinical practice guidelines have recommended structured diagnostic work-up in ARDS patients. In the LUNG SAFE study, lung CT-scan, bronchoalveolar lavage or auto-immunity tests were seldom performed, even in ARDS patients without documented diagnostic [8]. In the absence of obvious diagnosis and in case of non-documented pneumonia, we propose here a second-line diagnostic work-up (Table 2).
Reviewer 2 Report
Summary
This review questions the current Berlin definition for ARDS diagnosis in terms of its weaknesses and applicability considering the COVID-19 pandemic in ARDS non-invasively ventilated patients. This revision approaches a current dilemma, related to the need of expanding and reevaluating the criteria to diagnose ARDS in patients that are breathing spontaneously with noninvasive ventilation. However, the authors concern are not original, it echoes many other already published manuscripts that present this dilemma since the COVID pandemic started (PMC7202792, PMC7912364, PMC8289635, PMC8380483, PMC7160817, PMC7373003).
The authors approach very well the current limitations of the diagnosis criteria of ARDS, especially in mild cases that are frequently unrecognized by clinicians. In terms of ARDS severity, the authors question the validity of on assessing severity with a single blood gas measurement without standardized ventilator setting such as PEEP. Furthermore, this last setting cannot be considered for those with high-flow nasal oxygen only.
General concept comments
Review:
Specific comments
· The manuscript is relevant for the field but lacks to defend how expanding ADRS definitions to non-ventilated patients will correlate with an improved treatment.
· It requires to comprehensive re-write of the abstract. The second sentence in the abstract includes a question mark that seems to be out of context.
The cited references are current, nonetheless, it should incorporate additional references that have been recently published related to the same topic.
In general, this manuscript has poor figures and tables, they do not reflect the problematic of not having ARDS criteria expanded, which is the topic in the manuscript.
Table 2 is poorly designed and difficult to understand. It needs to be improved and clarify what BAL and Sputum are considered under the same category, why autoimmune and hypersensitivity are only listed under serum.
Figure 1. needs improvement. It only contemplates two lines of diagnosis work up, one for Community acquired pneumonias but the second doesn’t define diagnosis specific.
General questions
The review is clear in terms of approaching the deficiencies present in the criteria for ARDS in patients breathing spontaneously.
This review has limitations that need to be addressed by the authors. One is limited in explaining how expanding the ARDS criteria to spontaneously breathing subject will be beneficial specially for COVID-19 subjects where most of the decisions evolve around oxygen levels to decide intubation or not.
As noted above similar review published recently and, despite being of interest for the medical community it lacks specific relevance for patient management, which most of the time requires unique customized-patient treatment.
Conclusions fail to address why ARDS definition expansion with early implementation of ARDS-specific therapy will benefit those spontaneously breathing or those on High-flow nasal oxygen. Concluding that a complete diagnostic work-up is needed for ARDS diagnosis is not innovative since this is part of the standard of care of many institutions.
Author Response
Reviewer 2
Summary
This review questions the current Berlin definition for ARDS diagnosis in terms of its weaknesses and applicability considering the COVID-19 pandemic in ARDS non-invasively ventilated patients. This revision approaches a current dilemma, related to the need of expanding and reevaluating the criteria to diagnose ARDS in patients that are breathing spontaneously with noninvasive ventilation. However, the authors concern is not original, it echoes many other already published manuscripts that present this dilemma since the COVID pandemic started (PMC7202792, PMC7912364, PMC8289635, PMC8380483, PMC7160817, PMC7373003).
Response:
Although the reviewer is entitled to think that our review is not original, the references to which you refer focus mainly knowing whether COVID19 ARDS is a “classical” ARDS in terms of histopathologic features and respiratory mechanical properties as compared to the others, and none of these references discuss the definition of ARDS in patients breathing spontaneously as we do herein. Here, the topic is completely different, as we focus on early recognition of ARDS regardless of the underlying cause. Therefore, we believe that the references used by the reviewer to suggest that our manuscript is not original do not seem adequate.
The authors approach very well the current limitations of the diagnosis criteria of ARDS, especially in mild cases that are frequently unrecognized by clinicians. In terms of ARDS severity, the authors question the validity of on assessing severity with a single blood gas measurement without standardized ventilator setting such as PEEP. Furthermore, this last setting cannot be considered for those with high-flow nasal oxygen only.
Response:
You are right and, according to the Berlin definition, severity is defined only in intubated patients (while patients under NIV are only classified as mild ARDS regardless of their PaO2/FiO2 ratio). Assessment of severity after PEEP optimization cannot be performed in patients treated with high-flow nasal oxygen, but neither in those treated with NIV. Although optimization of PEEP level before assessment of oxygenation is not possible in patients breathing spontaneously with noninvasive oxygenation strategies such as high-flow nasal oxygen or even NIV, 2 blood gases separated by a few hours would probably be necessary to confirm ARDS in this setting. However, while our review highlights the importance of early definition of ARDS, we cannot definitely determine a new definition, which would need to be established in a consensual manner by experts.
General concept comments
Review:
Specific comments
The manuscript is relevant for the field but lacks to defend how expanding ADRS definitions to non-ventilated patients will correlate with an improved treatment.
Response:
We changed one paragraph as follows to better explain why early identification of ARDS could be of paramount importance:
Early identification of ARDS is likely to be a major issue in assessment of future pharmacological and non-pharmacological treatments. Numerous anti-inflammatory drugs have been unsuccessfully assessed in ARDS. Steroids were initially evaluated at a late stage in the course of ARDS with the aim of preventing evolution toward fibrosis [52–54]. However, mortality was significantly higher when steroids were started after 2 weeks of evolution [54]. By contrast, several studies in which steroids were started early in the course of ARDS, i.e. within the first 3 days after intubation, have shown beneficial effects on outcomes with a decreased risk of death [55,56]. During the Covid-19 pandemic, it was shown that steroids significantly reduced the risk of death not only in patients under invasive mechanical ventilation, but also in patients breathing spontaneously under standard oxygen [57]. COVID-19 is characterized by a marked systemic inflammatory response and in this setting, steroids may be particularly effective. However, it cannot be ruled out that steroids could be beneficial in all forms of ARDS when initiated at an early stage [58]. Early identification of ARDS would allow assessment of anti-inflammatory drugs or future pharmacological treatments at a much earlier stage of the disease. Recognition of ARDS at an early stage might also be of paramount importance to initiate lung-protective measures in patients breathing spontaneously with a high respiratory drive. Indeed, an emerging concept, known as patient self-inflicted lung injury, deals with the risk of progression of lung injury in patients with respiratory failure generating large tidal volumes and subsequent high transpulmonary pressures [59]. Similarly, prone positioning has demonstrated its efficacy in ARDS treatment amongst invasively ventilated patients [60], and was recently proposed, with promising results regarding the risk of intubation in COVID-19 patients undergoing high-flow nasal oxygen [61]. Therefore, early identification of ARDS in patients breathing spontaneously and not yet intubated could be useful as well for early initiation of anti-inflammatory drugs as for lung-protective measures.
We also added in the abstract:
Early identification of ARDS in patients breathing spontaneously may allow assessment of earlier initiation of pharmacological and non-pharmacological treatments. In the same way, accurate identification of the ARDS etiology is obviously of paramount importance for early initiation of adequate treatment. The precise underlying etiological diagnostic (bacterial, viral, fungal, immune, malignant, drug-induced, etc…) as well as the diagnostic approach have been understudied in the literature. To date, no clinical practice guidelines have recommended structured diagnostic work-up in ARDS patients. In addition to lung-protective ventilation with the aim of preventing worsening lung injury, specific treatment of the underlying cause is of paramount importance to improve outcomes. In this review, we discuss early identification of ARDS in non-intubated patients breathing spontaneously and propose a structured diagnosis work-up.
It requires to comprehensive re-write of the abstract. The second sentence in the abstract includes a question mark that seems to be out of context.
Response:
Some sentences had been cut when putting the text in form and the question mark was a typo error. We’re sorry for this mistake. The abstract has been rewritten.
The cited references are current, nonetheless, it should incorporate additional references that have been recently published related to the same topic.
Response:
I am very sorry but we cannot know what the reviewer is referring in to his/her comment. About which reference are you thinking??
In general, this manuscript has poor figures and tables, they do not reflect the problematic of not having ARDS criteria expanded, which is the topic in the manuscript. Table 2 is poorly designed and difficult to understand. It needs to be improved and clarify what BAL and Sputum are considered under the same category, why autoimmune and hypersensitivity are only listed under serum.
Response:
Table 2 has been redesigned to make it clearer: the left column presents microbiology tests, the right column presents auto-immunity/hypersensitivity tests, and each column is divided according to the considered biological sample. While there was no specific biological finding of auto-immune/hypersensitivity pathology in sputum, some BAL cytology results can provide valuable information.
Figure 1. needs improvement. It only contemplates two lines of diagnosis work up, one for Community acquired pneumonias but the second doesn’t define diagnosis specific.
Response:
We completely changed the figure to better represent the diagnostic work-up in the text.
General questions
The review is clear in terms of approaching the deficiencies present in the criteria for ARDS in patients breathing spontaneously.
This review has limitations that need to be addressed by the authors. One is limited in explaining how expanding the ARDS criteria to spontaneously breathing subject will be beneficial specially for COVID-19 subjects where most of the decisions evolve around oxygen levels to decide intubation or not.
Response:
As above-mentioned, we have rewritten the paragraph to explain how expanding the ARDS criteria to spontaneously breathing subject could be beneficial for ARDS.
Early identification of ARDS is likely to be a major issue in assessment of future pharmacological and non-pharmacological treatments. Numerous anti-inflammatory drugs have been unsuccessfully assessed in ARDS. Steroids were initially evaluated at a late stage in the course of ARDS with the aim of preventing evolution toward fibrosis [52–54]. However, mortality was significantly higher when steroids were started after 2 weeks of evolution [54]. By contrast, several studies in which steroids were started early in the course of ARDS, i.e. within the first 3 days after intubation, have shown beneficial effects on outcomes with a decreased risk of death [55,56]. During the Covid-19 pandemic, it was shown that steroids significantly reduced the risk of death not only in patients under invasive mechanical ventilation, but also in patients breathing spontaneously under standard oxygen [57]. COVID-19 is characterized by a marked systemic inflammatory response and in this setting, steroids may be particularly effective. However, it cannot be ruled out that steroids could be beneficial in all forms of ARDS when initiated at an early stage [58]. Early identification of ARDS would allow assessment of anti-inflammatory drugs or future pharmacological treatments at a much earlier stage of the disease. Recognition of ARDS at an early stage might also be of paramount importance to initiate lung-protective measures in patients breathing spontaneously with a high respiratory drive. Indeed, an emerging concept, known as patient self-inflicted lung injury, deals with the risk of progression of lung injury in patients with respiratory failure generating large tidal volumes and subsequent high transpulmonary pressures [59]. Similarly, prone positioning has demonstrated its efficacy in ARDS treatment amongst invasively ventilated patients [60], and was recently proposed, with promising results regarding the risk of intubation in COVID-19 patients undergoing high-flow nasal oxygen [61]. Therefore, early identification of ARDS in patients breathing spontaneously and not yet intubated could be useful as well for early initiation of anti-inflammatory drugs as for lung-protective measures.
As noted above similar review published recently and, despite being of interest for the medical community it lacks specific relevance for patient management, which most of the time requires unique customized-patient treatment.
Conclusions fail to address why ARDS definition expansion with early implementation of ARDS-specific therapy will benefit those spontaneously breathing or those on High-flow nasal oxygen.
Response:
We believe that earlier recognition of ARDS is a promising starting point insofar as no ARDS-specific intervention has demonstrated any benefit in these patients. Earlier recognition, especially in COVID-19 patients, could help to assess early intervention. The review focuses on earlier recognition and precise diagnosis of ARDS more than patient-centered treatment
Concluding that a complete diagnostic work-up is needed for ARDS diagnosis is not innovative since this is part of the standard of care of many institutions.
Response:
With all due respect, we fully disagree with the reviewer, and the LUNG SAFE study well demonstrates that diagnostic work-up is clearly not a standard of care of many institutions given that a high proportion of ARDS remains without diagnostic and that CT scan and bronchoalveolar lavage are seldom performed. Lack of precise etiological diagnosis in the largest study to date on ARDS (LUNG SAFE) seems to be a good example of the need for a consensual diagnosis work-up.
We added in the text the following:
To date, no clinical practice guidelines have recommended structured diagnostic work-up in ARDS patients. In the LUNG SAFE study, lung CT-scan, bronchoalveolar lavage or auto-immunity tests were seldom performed, even in ARDS patients without documented diagnostic [8]. In the absence of obvious diagnosis and in case of non-documented pneumonia, we propose here a second-line diagnostic work-up (Table 2).
Moreover, we indicate at the end of the manuscript the following, illustrating the point that diagnostic work-up is clearly not a standard of care of many institutions: After compete diagnostic work-up in all ARDS mimickers, Gibelin and colleagues showed that more than 75% of patients had a specific diagnosis [75]. Contrary to this study, the LUNG-SAFE study revealed that specific diagnosis was made in less than 20% of cases of ARDS mimickers [9]. However, lung CT-scan was performed in fewer than one-third of cases, bronchoalveolar lavage in fewer s than 10%, and auto-immunity in only 5%, there-by underlining the need for a complete diagnostic work-up in this population.
Round 2
Reviewer 2 Report
- Summary
The manuscript entitled “Early identification and diagnostic approach in acute respiratory distress syndrome (ARDS)”. This review examines the current Berlin definition for ARDS diagnosis in terms of its weaknesses and applicability considering the COVID-19 pandemic in ARDS non-invasively ventilated patients. This review approach very well the current limitations of the ARDS Berlin criteria and focused on the importance of early diagnosing of mild cases.
This revision approaches a current dilemma, related to the need of expanding and reevaluating the criteria to diagnose ARDS in patients that are breathing spontaneously with noninvasive ventilation.
The authors have rewritten the abstract, adding important information and in a more comprehensible format. Additionally, they have incorporated a valuable support to their arguments based on the LUNG SAFE study report.
Review:
- Specific comments
- The manuscript is relevant for the field. However, the title states that this review is focused on early identification and diagnostic approach in ARDS; indication section 3, as the most relevant section on the manuscript, “Criteria for ARDS in patients breathing spontaneously”. Please consider strengthen this section by presenting a clearer approach related to the authors proposed diagnostic criteria suggested for the early identification of this ARDS subpopulation. Section 3 will benefit from a diagram presenting the criteria mentioned in the subheading of this section. The criteria proposed by the authors are vaguely mentioned.
Section 4 presents the approach for identifying the etiology of ARDS but not for diagnosis ARDS. Please modify this subheading.
Author Response
Review:
Specific comments
The manuscript is relevant for the field. However, the title states that this review is focused on early identification and diagnostic approach in ARDS; indication section 3, as the most relevant section on the manuscript, “Criteria for ARDS in patients breathing spontaneously”. Please consider strengthen this section by presenting a clearer approach related to the authors proposed diagnostic criteria suggested for the early identification of this ARDS subpopulation. Section 3 will benefit from a diagram presenting the criteria mentioned in the subheading of this section. The criteria proposed by the authors are vaguely mentioned.
Response: We believe our review cannot stated a new definition of ARDS. Such new definition must be stated by an expert panel with consensus. We believe that the reviewer goes too far in these proposed changes. Our review allows to discuss new ideas for the future, not make a recommendation. Therefore, we conclude the section 3 by the following new sentence: We believe that the future definition could consider patients treated with at least 30 L/min of high-flow oxygen or with noninvasive ventilation as having actually ARDS and allow to consider severity as mild, moderate or severe ARDS according their PaO2/FiO2 measured under this noninvasive oxygenation strategy, and not only as mild ARDS.
Section 4 presents the approach for identifying the etiology of ARDS but not for diagnosis ARDS. Please modify this subheading.
Response: The reviewer’s comment is not clear. Yes, section 4 presents the approach for identifying the etiology of ARDS, not for early diagnosis ARDS which is discussed section 3. We definitely don’t agree with the reviewer who considers that only early diagnosis of ARDS is a crucial point. Our review focuses on early identification and diagnostic approach in acute respiratory distress syndrome (ARDS) as indicated in the title. To make it clearer we changed the title of subheading in the section 3 as following: 3. Early identification of ARDS in patients breathing spontaneously, and section 4. Proposed diagnostic approach for identification of ARDS etiology